# Ranking Items in Large-Scale Item Search Engines with Reinforcement Learning

**Chang GAO**

Department of Systems Engineering and Engineering Management
The Chinese University of Hong Kong
Shatin, Hong Kong
gaochang@se.cuhk.edu.hk

## Abstract

Ranking items in large-scale item search engines such as Amazon and Taobao is a typical multi-step decision-making problem. Due to the interactive nature between the human user and the search engine, reinforcement learning is a natural solution to this problem. In this project, we use Virtual-Taobao as the environment and some effective methods to solve this problem. Experimental results show that TD3 performs best on this problem. The video link is https://cuhk.zoom.us/rec/share/2XwILIEaNmJ0k_i_ujmtBqOZebJdz761-o1T3dlwe1uqgbvfUIohYKV_7Yms3h9y.BGkA7KAPrk2_aXVv Passcode: *tji34.7

## 1   Introduction

In E-commerce platforms such as Amazon and TaoBao, commodity search is the fundamental infrastructure, affording users the opportunities to search commodities, browse product information and make comparisons. Ranking items in a search session is a typical multi-step decision-making problem. Learning to rank (LTR) methods have been widely applied to ranking problems. However, such methods often consider different ranking steps in a session to be independent, which conversely may be highly correlated to each other. Due to the interactive nature between the human user and the search engine, reinforcement learning (RL) is a natural solution to this problem.

## 2   Related Work

### 2.1   Ranking Items in Large-Scale Item Search Engines

Hu et al. (2018) formulate the multi-step ranking problem as a search session Markov decision process (SSMDP). Then they propose the DPG-FBE algorithm to solve this problem. Afterward, they conduct two groups of experiments: a simulated experiment in which they construct an online shopping simulator and test the DPG-FBE algorithm and a real application in which they apply the DPG-FBE algorithm in TaoBao, one of the largest E-commerce platforms in the world. Shi et al. (2019) build a simulator named Virtual-Taobao, which is trained from the real-data of TaoBao. Compared with the real Taobao, Virtual-Taobao faithfully recovers important properties of the real environment.

### 2.2   Reinforcement Learning

Reinforcement Learning has made significant progress in recent years. "Deep Q Network" (DQN) (Mnih et al., 2015) algorithm is capable of human level performance on many Atari video games using unprocessed pixels for input. However, it can only handle discrete and low-dimensional action

spaces. Many tasks of interest have continuous and high dimensional action spaces. Some effective methods have been proposed to tackle continuous control problems, such as TRPO (Schulman et al., 2015), PPO (Schulman et al., 2017), DDPG (Lillicrap et al., 2015), TD3 (Fujimoto et al., 2018) and Soft Actor-Critic (SAC) (Haarnoja et al., 2018). TRPO and PPO are on-policy methods, while DDPG, TD3 and SAC are off-policy methods.

## 3 Problem description

In this project, we focus on the problem of ranking items in large-scale item search engines, which refers to assigning each item a score and sorting the items according to their scores. Generally, a search session between a user and the search engine is a multi-step ranking problem as follows:

(1) The user inputs a query in the blank of the search engine.

(2) The search engine ranks the items related to the query and displays the top $K$ items (e.g., $K = 10$) in a page.

(3) The user makes some operations (e.g., click items, buy some certain item or just request a new page of the same query) on the page.

(4) When a new page is requested, the search engine reranks the rest of the items and display the top $K$ items.

These four steps will repeat until the user buys some items or just leaves the search session. Empirically, a successful transaction always involves multiple rounds of the above process.

The multi-step ranking problem can be solved with a RL framework. The users act as the environment, while the search engine is the agent. The RL ranking framework consists of two major components. The first one is the simulator, which is learned from historical user behavior data and serves as the virtual environment. The second one is the RL Ranker. The RL Ranker interacts with the learned virtual environment, which can be formulated as follows:

Let $S = \{q, p_1, p_2, \ldots, p_n\}$ denote a search session, where $q$ is the query, $p_i$ is the $i$th page returned by the search engine, and $n$ is the number of pages in the session. At each time step, the user can continue the search session by requesting a new page or terminate the search session by purchasing an item or leaving the search session. The user behavior will be influenced by what he/she wants and what he/she sees, which will be reflected by $\mathcal{O}$ and $\mathcal{A}$, in which $\mathcal{O}$ is the engine observation, i.e. user feature with request, and $\mathcal{A}$ is the engine action, i.e. the displayed pages. For the engine, if the user clicks on an item, a reward of 1 will be given. Otherwise, the reward will be 0. When training the RL Ranker, the parameters in the simulator are fixed and the parameters in the RL Ranker are updated by the reward signals.

## 4 Environment

We use Virtual-Taobao (Shi et al., 2019) as the environment. Virtual-Taobao simulates the customers, items, and recommendation system. An interactive process between the system and a customer is as follows:

(1) Virtual-Taobao samples a feature vector of the customer, including both the customer's description and customer's query.

(2) The system retrieves a set of related items according to the query, forming the whole itemset.

(3) The system uses a model to assign a weight vector corresponding to the item attributes.

(4) The system calculates the product between the weight vector and the item attributes for each item, and selects the top 10 items with the highest values.

(5) The selected 10 items are pushed to the customer. Then, the customer will choose to click on some items (results in CTR++), browse the next page (results in starting over from step 2 with changed customer features), or leave the platform (results in the end of the session).

In the above process, the model in step (3) is to be trained. The model inputs the features of the customer and the itemset, and outputs a weight vector.

# 5 Experiments

The multi-step ranking problem is a continuous task. We use the following methods to solve this problem on the Virtual-Taobao environment: vanilla policy gradient (PG), PPO, DDPG, TD3 and SAC.

## 5.1 Results

We use different hyperparameters to train the models and report the best results. The main metric is click-through rate (CTR). The higher the CTR, the better the performance of the model. Every time a user clicks on an item, there is a reward 1. Thus the reward depends on the length of the episode which depends on the user. CTR dose not have this problem.

Figure 1 and Figure 2 show the training CTR and test CTR of five different models: TD3, PPO, SAC, DDPG and PG, respectively. We can see that TD3 performs better than other methods and can obtain very good performance when trained with only 200k steps. PPO and SAC have similar performance. However, the learning curve of PPO is more stable. DDPG can obtain a good performance in a few steps, but it will get stuck afterwards. PG performs the worst, but can achieve performance comparable to DDPG after training more steps. In addition, we find that SAC is less sensitive to hyperparameters compared with other models.

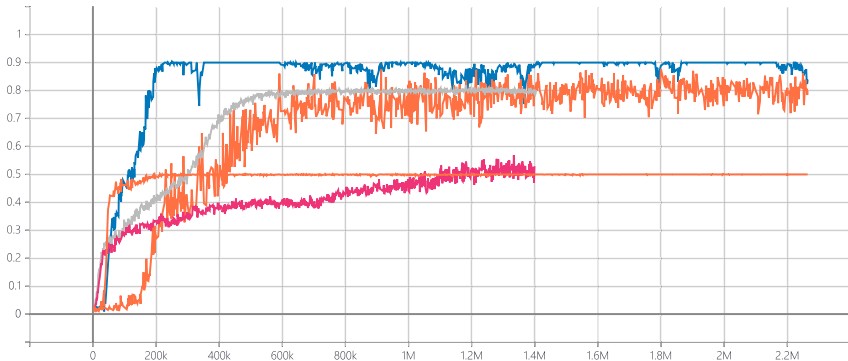

Figure 1: Training CTR of five models: TD3 (blue), PPO (grey), SAC (orange above), DDPG (orange below) and PG (pink).

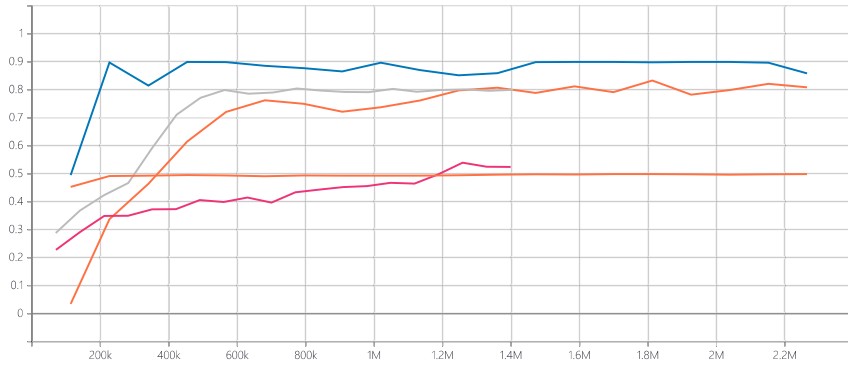

Figure 2: Test CTR of five models: TD3 (blue), PPO (grey), SAC (orange above), DDPG (orange below) and PG (pink).

When training PPO, we find that some tricks have a great impact on performance. We have tried two important tricks: value clip (Ilyas et al., 2018) and reward normalization. Figure 3 shows the training CTR of three versions of PPO. We can see that both tricks can improve the performance of PPO.

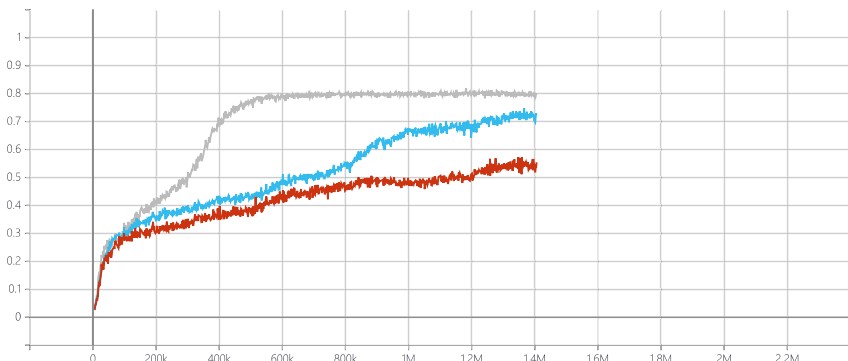

Figure 3: Training CTR of PPO (red), PPO with value clip (blue) and PPO with value clip and reward normalization (grey).

## 6  Conclusion

In this project, we use Virtual-Taobao as the environment and use different methods to solve the ranking problem. Experimental results show that TD3 performs better than other methods and some tricks such as reward normalization can greatly affect the performance.

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
