# OpenReview forum: "Ranking Items in Large-Scale Item Search Engines with Reinforcement Learning"
_CUHK.edu.hk/2021/Course/IERG5350_

### Official Review · AnonReviewer2 · 2020-12-15
**Accept as the paper provides a nice implementation of the state-of-the-art algorithms but lacks innovative ideas**

**Rating:** 7
**Confidence:** 5

**Review:**

In this project, the author customizes a reinforcement learning agent to rank items for commodity search engines. Specifically, the open-source Virtual-Taobao environment is used, where an RL agent learns to assign weights corresponding to different item attributes and the simulated customer will make reactions such as clicking on the product and leaving the platform. Some of the state-of-the-art algorithms, i.e., vanilla PG, PPO, DDPG, TD3 and SAC are evaluated for this task, and some tricks are implemented in PPO to improve the results. In general, I think it is a nice practice to try existing algorithms, but there is a lack of innovation in the idea or implementation.

Here are some comments for the author:

1.The problem description and task specification are clear. The introduction of related work is easy to understand.

2.The paper is well-written and easy to read. The organization of content is appropriate. However, some details need to be paid attention to. For example, the word “CTR” firstly appears in Section 4 without being explained. Also, there lacks a detailed definition of the click-through rate in Section 5. Perhaps it is well-known in this reseach topic, but the authors are suggested to include the full definition of the term for readers outside this field.

3.The experimental results are clearly illuatrated and described, and the performance of PPO with different tricks (value clip and reward normalization) are discussed.

4.In terms of the method, it is a nice practice to try the state-of-the-art RL algorithms in this task, but there is hardly any innovative ideas or implementation, though. In terms of the environment implementation, the Virtual-Taobao environment is open-source and designed for RL usage. Therefore, I think the contribution of this work is minor.
5.
In summary, my opinions about this work is concluded in terms of these four aspects:

- **Orininality: 6** (The paper investigates the state-of-the-art reinforcement learning algorithms for item ranking. Different tricks are implemented but the ideas are not very new. )

- **Quality: 7** (The authors have done a set of experiments to compare different methods for the target application. A detailed analysis of the results is provided.)

- **Clarity: 8** (The paper is well-written, which clearly defines the problem and describes the experimental resuls in detail.)

- **Significance: 7** (Learning to rank items for commodity search engines is a hot topic in both research and industry. Investigating the state-of-the-art reinforcement learning algorithms in this application will help develop better data mining methods.)

I think the work done in this project satisfies the requirement of the course in general.

---

### Official Review · AnonReviewer1 · 2020-12-16
**a well defined problem and good application of exist methods**

**Rating:** 8
**Confidence:** 4

**Review:**

General:
The paper tried to use reinforcement learning to solve ranking large-scale items problem in search engines. This experiment based on Virtual-Taobao, and the system will retrives a set items according to the query requested by the customer. We can regard it as a successful search when  system the customer click on the item and buy it, and reinforcement learning seems to be a natural solution to this problem. Finally, the author also use several methods to solve this problem: vanilla policy gradient(PG), PPO, DDPG, TD3 and SAC and compare the results.

Pros:
1. The task is clearly defined, and this paper Is written logically
2. The experiments of this paper are enough, and apply several methods to this task so that we can compare the result.
3. According to this paper, it seems that the multi-step ranking problem is a quite novel topic, it may be better if the author can  compare it with traditional task.

Cons:
1. Lack of further analysis of this result, for example, the reason that why some method can perform better than the other.
2. Maybe need more innovation.

---

### Official Review · AnonReviewer3 · 2020-12-18
**Well written project paper but lacks detailed discussion and novelty**

**Rating:** 6
**Confidence:** 5

**Review:**

In this paper, the author chooses Virtual-Taobao as the environment and uses different RL algorithms to solve the ranking problem. The paper is well organized and described.
There are some comments for this paper,
1. the project is well described and organized.
2. The author conducted five RL algorithms, but rarely discusses the challenges of applying these algorithms to ranking problems.
3. The discussion on the results is less.
4. Any comparison with LTR methods?